# High Fidelity Text-Guided Music Editing via Single-Stage Flow Matching

## Abstract

We introduce MELODYFLOW, an efficient text-controllable high-fidelity music generation and editing model. It operates on continuous latent representations from a low frame rate 48 kHz stereo variational auto encoder codec. Based on a diffusion transformer architecture trained on a flow-matching objective the model can edit diverse high quality stereo samples of variable duration, with simple text descriptions. We adapt the RENOISE latent inversion method to flow matching and compare it with the original implementation and naive denoising diffusion implicit model (DDIM) inversion on a variety of music editing prompts. Our results indicate that the regularized latent inversion outperforms both RENOISE and DDIM for zero-shot test-time text-guided editing on several objective metrics. Subjective evaluations exhibit a noticeable improvement over previous state of the art for music editing. Code and model weights will be publicly made available. Samples are available at `https://melodyflow.github.io`.

## 1 Introduction

Text-conditioned music generation has made tremendous progress in the past two years (Schneider et al., 2023; Huang et al., 2023; Agostinelli et al., 2023; Copet et al., 2024; Ziv et al., 2023; Liu et al., 2023b; Li et al., 2023; Prajwal et al., 2024). The prevailing method for audio representation involves compressing the waveform into a series of discrete or continuous tokens, and then training a generative model on top of those. Two dominant generative model architectures have emerged, one based on autoregressive Language Models (LMs) (Agostinelli et al., 2023; Copet et al., 2024), the other on diffusion (Schneider et al., 2023; Huang et al., 2023; Liu et al., 2023b; Li et al., 2023; Prajwal et al., 2024). A third method sometimes referred to as discrete diffusion relies on non-autoregressive masked token prediction (Ziv et al., 2023; Garcia et al., 2023). The target level of audio fidelity depends on the models and some have already successfully generated 44.1 kHz or high stereo signals (Schneider et al., 2023; Li et al., 2023; Evans et al., 2024a).

The increasing popularity of diffusion models in computer vision has led to the emergence of a new area of research focused on text-controlled audio editing (Wang et al., 2023; Lin et al., 2024; Garcia et al., 2023; Wu et al., 2023; Novack et al., 2024; Zhang et al., 2024; Manor & Michaeli, 2024). The sound design process often involves multiple iterations, and using efficient editing methods is a key approach to achieving this effectively. Music editing encompasses a wide range of tasks, including but not limited to: inpainting/outpainting, looping, instrument or genre swapping, vocals removal, lyrics editing, tempo control, and recording conditions modification (e.g. from studio quality to a concert setting). Recent works have addressed some of these tasks using specialized models (Wang et al., 2023; Garcia et al., 2023; Lin et al., 2024; Wu et al., 2023; Copet et al., 2024) or zero-shot editing methods from the computer vision domain, which are exclusive to diffusion models (Novack et al., 2024; Zhang et al., 2024; Manor & Michaeli, 2024). Despite recent efforts, no approach has yet shown the ability to perform high-fidelity generic style transfer across various music editing tasks. This limitation can be attributed to several factors, including insufficient high-quality data, inadequate foundational music generation models, and design choices that fail to generalize effectively to diverse editing tasks. Inference speed is crucial for creatives, and the music domain presents a unique challenge due to the high-fidelity (48 kHz stereo) requirement in the sound design process. Recently Lipman et al. (2022) proposed the Flow Matching (FM) generative modeling formulation, which involves constructing optimal transport paths between data and noise samples. Flow Matching (FM) offers a more robust and stable approach to training diffusion models, with the added benefit of faster

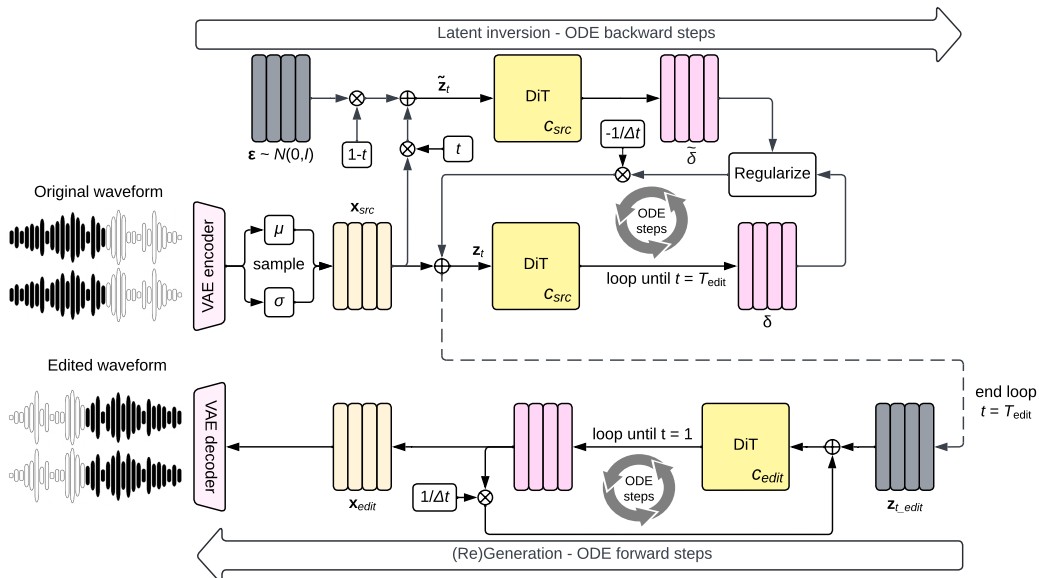

Figure 1: Overview of the MELODYFLOW editing process. A waveform is encoded into $\mathbf{x}_{src}$ before being fed to the ODE solver. Step-by-step, the DiT predicts the velocity $\delta$ from data to noise, while being regularized against the prediction of an artificially constructed $\tilde{\mathbf{z}}_t$ so as to enhance editability. Once the target inversion flow step $T_{edit}$ has been reached, the model is used in the classic generation setting (bottom of the Figure, from right to left), except that the starting latent $\mathbf{z}_{t_{edit}}$ has been estimated so as to achieve better editability and consistency with the source waveform.

inference. This method has been successfully applied to train foundational speech (Le et al., 2024) and audio (Vyas et al., 2023) generative models. For the music domain Prajwal et al. (2024) utilized a two-stage FM model for text-guided music generation, where the first stage generates semantic features and the second stage generates acoustic features.

In this work we present MELODYFLOW, a single-stage text-conditioned FM model designed for instrumental music generation and editing. The model operates on continuous representations of a low frame rate Variational Audio Encoder (VAE) codec. Additionally, thanks to the versatility of FM, MELODYFLOW is compatible with any zero-shot test-time editing method such as DDIM inversion (Song et al., 2020) or ReNoise (Garibi et al., 2024). We enhance the editability of the FM inversion by adapting the latent inversion of Garibi et al. (2024) to the FM formulation. Both our objective and subjective evaluations on music editing indicate that MELODYFLOW can support a diversity of editing tasks on real songs without any finetuning, achieving fast music editing with remarkable consistency, text-adherence and minimal quality loss compared with original samples. In addition we conduct an ablation study on the importance of the key design choices on the overall model quality/efficiency trade off.

**Our contributions:** (i) We introduce the first of its kind single-stage text-to-music FM model to generate and edit 48 kHz stereo samples of up to 30 seconds, with enhancements in both the audio latent representation and generative model, striking a better balance between quality and efficiency. (ii) We explore a novel regularized FM inversion method capable of performing faithful zero-shot test-time text-guided editing on various axes while maintaining coherence with the original sample. (iii) We publicly release the code and model weights to foster research on music editing.

## 2 METHOD

MELODYFLOW combines a continuous audio codec, a text-conditioned Diffusion Transformer (DiT) FM model and a regularized latent inversion method. The model can perform text-guided editing of real or generated audio samples. The overall editing process is depicted in the Figure 1.

## 2.1 LATENT AUDIO REPRESENTATION

Our codec derives from EnCodec (Défossez et al., 2022) with additional features from the Descript Audio Codec (DAC) (Kumar et al., 2024) (snake activations, band-wise STFT discriminators) and Evans et al. (2024a) (KL-regularized bottleneck, perceptual weighting). A convolutional auto-encoder encodes the waveform into a sequence of latent bottleneck representations, its frame rate function of the convolution strides. Audio fidelity is enforced by multi-scale STFT reconstruction losses complemented by the sum and difference STFT loss for stereo support (Steinmetz et al., 2020).

## 2.2 CONDITIONAL FLOW MATCHING MODEL

Given an audio sample $\mathbf{a} \in \mathbb{R}^{D \times f_s}$, a sequence $\mathbf{x} \in \mathbb{R}^{L \times d}$ of latent representations is extracted by the neural codec. FM models the optimal transport paths that map a sequence $\epsilon \in \mathbb{R}^{L \times d} \sim \mathcal{N}(0, I)$ to $\mathbf{x}$ trough a linear transformation - function of the flow step $t$ - following equation 2.2.

$$\mathbf{z}_t = t\mathbf{x} + (1 - t)\epsilon, t \in [0, 1]$$

During training, $t$ is randomly sampled and the DiT $\Theta$ is trained to estimate $d\mathbf{z}_t/dt$ conditioned on $t$ and a text description $c$.

$$d\mathbf{z}_t/dt = v_\Theta(\mathbf{z}_t, t, c) = \mathbf{x} - \epsilon$$

By design, after training, the model can be used with any ODE solver to estimate $\mathbf{x} = \mathbf{z_1}$ given $\epsilon = \mathbf{z_0}$ (and vice versa), and a text description. The text-to-music inference happens as such: starting from a random noise vector $\epsilon \in \mathbb{R}^{L \times d} \sim \mathcal{N}(0, I)$ and a text description $c$ of the expected audio the ODE solver is run from $t = 0$ to $t = 1$ to estimate the most likely sequence of latents $\mathbf{x}_{generated}$.

$$\mathbf{x}_{generated} = \mathbf{ODE}_{0 \rightarrow 1}(\epsilon, c)$$

After the latents have been estimated they are fed to the codec decoder to materialize the waveform. Kingma & Gao (2024) show that the flow step sampling density during training plays an important role in model performance. In our implementation $t$ is sampled from a logit-normal distribution (Karras et al., 2022; Esser et al., 2024).

## 2.3 TEXT-GUIDED EDITING THROUGH LATENT INVERSION

Due to the bijective nature of the FM formulation (where given a text condition, each latent sequence is mapped to a single noise vector), the model is compatible with existing latent inversion methods such as DDIM inversion (Song et al., 2020). Given the latent representation $\mathbf{x}_{src}$ of an existing audio with an optional accompanying caption $c \in \{\varnothing, c_{src}\}$, the model can estimate its corresponding noise (or intermediate) representation $\mathbf{z}_{t_{edit}} = \mathbf{ODE}_{t_{edit} \leftarrow 1}(\mathbf{x}_{src}, c)$ by running the ODE solver in the backward direction until an intermediary time step $t_{edit}$ (top of the Figure 1). Given the intermediary representation $\mathbf{z}_{t_{edit}}$, the ODE forward process can be conditioned on a new text description $c_{edit}$ that materialises the editing prompt: $\mathbf{x}_{edit} = \mathbf{ODE}_{t_{edit} \rightarrow 1}(\mathbf{z}_{t_{edit}}, c_{edit})$. A good inversion process should accurately reconstruct the input when $c_{edit} = c_{src}$, as shown in equation 2.3.

$$\mathbf{x}_{edit} = \mathbf{ODE}_{t_{edit} \rightarrow 1}(\mathbf{ODE}_{t_{edit} \leftarrow 1}(\mathbf{x}_{src}, c \in \{\varnothing, c_{src}\}), c_{src}) \approx \mathbf{x}_{src}$$

In such case when swapping $c_{src}$ for $c_{edit}$ in the $t_{edit} \rightarrow 1$ forward direction, the expectation is for the generated audio to preserve some consistency with the source while being faithful to the prompt. However in practice it was observed by Mokady et al. (2023) that DDIM inversion suffers from poor editability due to the classifier free guidance.

## 2.4 REGULARIZED LATENT INVERSION

Even though FM consists in estimating straight trajectories, in practice those are never completely straight and the edited samples do not preserve enough consistency with the source.

1. The distribution of predicted velocities tends to shift away from that of training due to the classifier free guidance (Mokady et al., 2023), which can lead to divergence of the inversion trajectory. This was observed by Parmar et al. (2023) with $\epsilon$-prediction, which they address by adding an autocorrelation regularization during inversion to preserve the statistical properties of the predictions.

2. Any pair of successive $(\mathbf{z}_t, \mathbf{z}_{t-\triangle t})$ along the inversion path usually has estimated velocities $v_\Theta(\mathbf{z}_t, t, c) \neq v_\Theta(\mathbf{z}_{t-\triangle t}, t - \triangle t, c)$, which affects reversibility (hence the consistency with the source sample). Building a fully reversible inversion path requires estimating $\mathbf{z}'_{t-\triangle t}$ such that $v_\Theta(\mathbf{z}_t, t, c) \approx v_\Theta(\mathbf{z}'_{t-\triangle t}, t - \triangle t, c)$, for example following Garibi et al. (2024).

RENOISE (Garibi et al., 2024) addresses those two problems by combining both $\epsilon$-prediction regularization and reversible inversion trajectory estimation. Applying RENOISE to FM requires either (1) reformulating FM as $\epsilon$-prediction or (2) adapting the regularization mechanism. Indeed since our FM model predicts the velocity $v_\Theta(\mathbf{z}_t, t, c) = \mathbf{x} - \epsilon$ and RENOISE operates on noise predictions, applying RENOISE to FM (1) requires subtracting the source latent $\mathbf{x}_{src}$ from $v_\Theta$ to try and isolate and regularize $\epsilon$ directly. However in such setting the inversion diverges when conditioning on text ($c_{src}$) and using CFG (appendix A.2.4), likely due to $v_\Theta(\mathbf{z}_t, t, c) - \mathbf{x}_{src}$ not properly removing the signal component of $\mathbf{x} - \epsilon$ at lower flow steps. To prevent this behavior we propose to (2) directly regularize the FM prediction using only the KL regularization from Garibi et al. (2024). An thorough comparison between the considered approaches can be found in the sections 4.3.1, 4.3.2 and 4.5.2.

The Algorithm 1 details our proposed inversion. Each iteration consists in estimating a reversible inversion point $\mathbf{z}_{t-\triangle t}$ from a source point $\mathbf{z}_t$ such that $v_\Theta(\mathbf{z}_{t-\triangle t}, t - \triangle t, c) \approx v_\Theta(\mathbf{z}_t, t, c)$. In such case the jump from $\mathbf{z}_t$ to $\mathbf{z}_{t-\triangle t}$ is considered reversible. This is done iteratively in $K$ steps following the convergence property of Garibi et al. (2024). During each of those steps, the model prediction is regularized against the prediction of an artificially constructed $\tilde{\mathbf{z}}_{t-\triangle t}$ (also shown in the Figure 1).

---

**Algorithm 1** Proposed regularized FM inversion

---

**Input:** Sequence of audio latents $\mathbf{x}$. Number of ODE backward steps $S$. Source text description $c \in \{\varnothing, c_{src}\}$. K regularization steps with weights $\{w_k\}_{k=1}^K$, KL regularization weight $\lambda_{KL}$.
**Output:** A noisy latent $\mathbf{z}_{T_{edit}}$ such that $\mathbf{ODE}_{T_{edit} \to 1}(\mathbf{z}_{T_{edit}}, c_{src}) \approx \mathbf{x}$.
  $\triangle t \leftarrow (1 - T_{edit})/S$
  **for** $t = 1, 1 - \triangle t, \ldots, T_{edit} + \triangle t$ **do**
    $\mathbf{z}_{t-\triangle t}^{(0)} \leftarrow \mathbf{z}_t$
    **for** $k = 1, \ldots, K$ **do**
      $\delta \leftarrow v_\Theta(\mathbf{z}_{t-\triangle t}^{(k-1)}, t - \triangle t, c)$
      **if** $w_k > 0$ **then**
        sample $\epsilon \sim \mathcal{N}(0, I)$
        $\tilde{\mathbf{z}}_{t-\triangle t}^{(k-1)} \leftarrow \mathbf{x}(t - \triangle t) + \epsilon(1 - (t - \triangle t))$
        $\tilde{\delta} \leftarrow v_\Theta(\tilde{\mathbf{z}}_{t-\triangle t}^{(k-1)}, t - \triangle t, c)$
        $\delta \leftarrow \delta - \lambda_{KL}\nabla_\delta \mathcal{L}_{patchKL}(\delta, \tilde{\delta})$
      **end if**
      $\mathbf{z}_{t-\triangle t}^{(k)} \leftarrow \mathbf{z}_t - \delta \triangle t$
    **end for**
    $\mathbf{z}_{t-\triangle t} \leftarrow \frac{\sum_{k=1}^K w_k \mathbf{z}_{t-\triangle t}^{(k)}}{\sum_{k=1}^K w_k}$
  **end for**
  **return** $\mathbf{z}_{T_{edit}}$

---

## 2.5 IMPROVING FLOW MATCHING FOR TEXT-TO-MUSIC GENERATION

### 2.5.1 CODEC BOTTLENECK

Recently Prajwal et al. (2024) trained a two-stage music FM model on continuous latent representations, but both the semantic and acoustic latent representations where trained with a discretization objective (HuBERT semantic features and RVQ-regularized codec). The concurrent work of Evans et al. (2024b) demonstrated long form music generation capabilities by using a KL-regularized bottleneck in their codec with a temporal downsampling as low as 21.5 Hz. However none of these works have carefully investigated the influence of the bottleneck regulariser on both music reconstruction and generation performance, all other things being equal. Indeed Rombach et al. (2022) - a seminal work on VAE for image generation - note that *LDMs trained in VQ-regularized latent spaces achieve better sample quality* than KL-regularized ones. Our ablation in section 4.4 leads to a different

conclusion. Using a KL-regularizer achieves indeed better music reconstruction and generation performance for a much lower frame rate, which is key for faster inference.

### 2.5.2 Minibatch coupling

Tong et al. (2023) and Pooladian et al. (2023) expanded over prior work on FM modeling by sampling pairs $(\mathbf{x}, \epsilon)$ from the joint distribution given the by the optimal transport plan between the data $\mathbf{X} = \{\mathbf{x}^{(i)}\}_{i=1}^{B}$ and noise $\mathbf{E} = \{\epsilon^{(i)}\}_{i=1}^{B}$ samples within a batch of size $B$. Essentially this translates into running the Hungarian algorithm so as to find the permutation matrix $\mathbf{P}$ that minimizes $||\mathbf{X} - \mathbf{PE}||_2^2$. They demonstrate it results in straighter optimal transport paths during inference (that are closer to the theoretical linear mapping assumption between noise and data samples) and consequently offers better quality-efficiency trade offs. We shed light on the importance of mini-batch coupling in sections 4.5.1 and 4.5.2 where we underline the overall benefit of our FM model design choices on both music generation and editing.

## 3 Experimental setup

### 3.1 Model

MELODYFLOW uses a DiT of sizes 400M (small) and 1B (medium) parameters with U-shaped skip connections Bao et al. (2023). The model is conditioned via cross attention on a T5 representation (Raffel et al., 2020) computed from the text description of the music. The model integrates a specific L-shaped self-attention mask meant to better generalize to different segment lengths during inference (appendix A.2.2). The flow step is injected following Hatamizadeh et al. (2023). Minibatch coupling is computed with `torch-linear-assignment`[1]. MELODYFLOW-small (resp. MELODYFLOW-medium) is trained on latent representation sequences of 32 kHz mono (resp. 48 kHz stereo) segments of 10 (resp. 30) seconds, encoded at 20 Hz frame rate (resp. 25 Hz). From the codec perspective the only difference between encoding mono or stereo waveform is the number of input (resp. output) channels for the first (resp. last) convolution of the encoder (resp. decoder): 1 for mono and 2 for stereo. The appendix A.2.1 specifically investigates the impact of encoding stereo instead of mono signals on both reconstruction and generation performance. More details regarding audio representation and FM model implementation and training are provided in the appendix A.1.

### 3.2 Generation and editing

For text-to-music generation we use the `midpoint` ODE solver from `torchdiffeq` with a step size of 0.03125. A classifier free guidance (CFG) of 4.0 is chosen after grid search (appendix A.2.4). For music editing we use the same configuration for DDIM inversion. For RENOISE and MELODYFLOW we use a longer step size of 0.04 to account for the additional forward passes induced by the reversible trajectory estimation. For DDIM inversion this gives a total of 64 inversion and 64 generation steps (e.g. forward passes through the DiT). For RENOISE and MELODYFLOW the inversion takes 25 steps (each of them requires 4 iterations for the reversibility estimation) and 25 forward steps, for a total of 125. In summary MELODYFLOW's inversion is run with $S = 25, K = 4, w_0 = w_1 = 0, w_2 = 2, w_3 = 3$ and $\lambda_{KL} = 0.2$.

### 3.3 Datasets

**Training** Our training dataset is made of 10K high-quality internal music tracks and the Shutter-Stock and Pond5 music collections with respectively 25K and 365K instrument-only music tracks, totalling into 20k hours. All datasets consist of full-length music sampled at 48 kHz stereo with meta-data composed of a textual description sometimes containing the genre, BPM and key. Descriptions are curated by removing frequent patterns that are unrelated to the music (such as URLs). For 32 kHz mono models the waveform is downsampled and the stereo channels are averaged.

**Evaluation** For the main text-to-music generation results we evaluate MELODYFLOW and prior work on the MusicCaps dataset (Agostinelli et al., 2023). We compute objective metrics for

---

[1] https://github.com/ivan-chai/torch-linear-assignment

Table 1: Comparison to baselines on text-guided high fidelity music editing of samples from the IN-DOMAIN test set, using LLM-assisted editing prompts.

| MODEL | METHOD | OVL. ↑ | REL. ↑ | CON. ↑ | AVG. ↑ |
|---|---|---|---|---|---|
| AUDIOLDM 2-music | DDPM inv. | 2.48±0.07 | 2.36±0.08 | **2.72**±0.09 | 2.52 |
| MUSICGEN-melody | Chroma cond. | 2.57±0.08 | 2.46±0.09 | 2.14±0.07 | 2.39 |
| MELODYFLOW-medium | Reg. inv. | **2.72**±0.08 | **2.72**±0.07 | 2.61±0.10 | **2.68** |

MELODYFLOW and report those from previous literature. Subjective evaluations are conducted on a subset of 198 examples from the genre-balanced set. For ablations we rely on an in-domain held out evaluation set different from that of Copet et al. (2024), made of 8377 tracks. The same in-domain tracks are used for objective editing evaluations. Subjective evaluations of edits are run on a subset of 181 higher fidelity samples from our in-domain test set with LLM-assisted designed prompts (more details in appendix A.1.3).

## 3.4 METRICS

We evaluate MELODYFLOW using both objective and subjective metrics following the evaluation protocol of Kreuk et al. (2022) and Copet et al. (2024) for generation. Reported objective metrics are the Fréchet Audio Distance (FAD) (Roblek et al., 2019) with VGGish embeddings (Hershey et al., 2017), the Kullback–Leibler divergence (KLD) with PASST audio encoder (Koutini et al., 2021) and CLAP[2] cosine similarity (Elizalde et al., 2023). For music editing evaluations we compute the average L2 distance between the original and edited latent sequences (LPAPS (Iashin & Rahtu, 2021)), $\text{FAD}_{edit}$ between the distribution of source and edited samples and $\text{CLAP}_{edit}$ between the edited audio and the editing prompt. Subjective evaluations relate to (i) overall quality (OVL), and (ii) relevance to the text input (REL), both using a Likert scale (from 1 to 5). Additionally for music editing evaluations we report (iii) editing consistency (CON). Raters were recruited using the Amazon Mechanical Turk platform and all samples were normalized to -14dB LUFS (Series, 2011). For stereo samples objective evaluation the signal is down mixed into mono prior to metrics computation. For subjective ratings we keep the original audio format generated by each model. A screenshot of the evaluation form is presented in appendix A.1.4.

## 4 RESULTS

### 4.1 TEXT-GUIDED MUSIC EDITING

We compare MELODYFLOW-medium with existing open source music editing implementations, namely MUSICGEN-melody and AUDIOLDM 2 with DDPM inversion (following Manor & Michaeli (2024)). The Table 1 presents the main music editing subjective evaluation results. MELODYFLOW outperforms both baselines on the quality and text-fidelity axes. MUSICGEN-melody specifically underperforms consistency-wise while AUDIOLDM 2 suffers from lower text adherence. Indeed during our listening tests we observe that AUDIOLDM 2 with DDPM inversion sometimes only generates a distorted version of the original track, hence does not take into account the editing prompt and keeps a strong similarity with the original. This also explains why consistency-wise MELODYFLOW lags slightly behind AUDIOLDM 2. Averaging on the three axes MELODYFLOW sets a new baseline for zero-shot music editing at test-time.

### 4.2 TEXT-TO-MUSIC GENERATION

Text-to-music generation performance is reported in the Table 2. For text-to-music qualitative evaluations we compare MELODYFLOW to three baselines that also support both generation and editing: MUSICGEN, AUDIOLDM 2, STABLE-AUDIO. For MUSICGEN and AUDIOLDM 2 we use the available open source implementations and for STABLE-AUDIO we use the public API (as of Wed. May 14 2024, AudioSpark 2.0 model version). MELODYFLOW achieves comparable performance

---

[2]`https://github.com/LAION-AI/CLAP`

Table 2: Comparison to text-to-music baselines. We report the original objective metrics for AUDI-OLDM 2 and MUSICGEN. For subjective evaluations we report mean and CI95.

| MODEL | FAD$_{vgg}$ ↓ | KL ↓ | CLAP$_{sim}$ ↑ | OVL. ↑ | REL. ↑ | # STEPS | LATENCY (S) |
|---|---|---|---|---|---|---|---|
| Reference | - | - | - | 3.67±0.10 | 4.04±0.10 | - | - |
| AUDIOLDM 2 | 3.1 | 1.20 | 0.31 | 2.79±0.08 | 3.40±0.08 | 208 | 18.1 |
| MUSICGEN-small | 3.1 | 1.29 | 0.31 | - | - | 1500 | 17.6 |
| MUSICGEN-medium | 3.4 | 1.23 | 0.32 | 3.40±0.08 | 3.79±0.07 | 1500 | 41.3 |
| STABLE-AUDIO | - | - | - | 3.67±0.08 | 3.89±0.07 | 100 | 8.0 |
| MAGNET-small | 3.3 | 1.12 | 0.31 | - | - | 180 | 4.0 |
| MAGNET-large | 4.0 | 1.15 | 0.29 | - | - | 180 | 12.6 |
| MELODYFLOW-small | 2.8 | 1.27 | 0.33 | 3.27±0.08 | 3.83±0.08 | 64 | 1.8 |
| MELODYFLOW-medium | 3.5 | 1.30 | 0.31 | 3.41±0.08 | 3.77±0.07 | 64 | 2.3 |

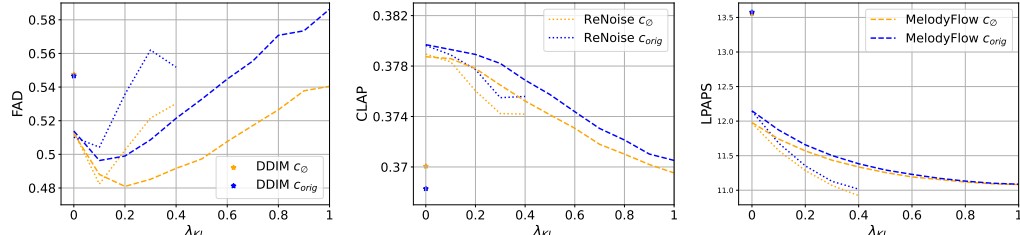

(a) FAD$_{edit}$ as a function of $\lambda_{KL}$.  (b) CLAP$_{edit}$ as a function of $\lambda_{KL}$.  (c) LPAPS as a function of $\lambda_{KL}$.

Figure 2: Effect of the regularization weight $\lambda_{KL}$ on the quality (Figure 2a) and text-adherence (Figure 2b) of music editing. $\epsilon$- and $v$-prediction are compared with or without $c_{orig}$.

with MUSICGEN, both lagging slightly behind STABLE-AUDIO in terms of human preference. We do not report objective metrics on STABLE-AUDIO as none were reported on the full MusicCaps benchmark Evans et al. (2024a). We do not run any subjective evaluation against MAGNET but report their objective metrics and latency values. MELODYFLOW achieves remarkable efficiency with only 64 inference steps.

### 4.3 LATENT INVERSION

#### 4.3.1 INVERSION METHODS

We compare MELODYFLOW with DDIM and RENOISE in the Figures 2a, 2b and 2c, as a function of the divergence loss weight $\lambda_{KL}$. During the inversion we use a classifier-free-guidance (CFG) of 0 and employ a CFG of 4 during the regeneration. The choice of zero CFG is meant to prevent divergence during inversion (see A.2.4). For RENOISE and MELODYFLOW the predictions are regularized by the weighted KL patch-wise divergence loss $\mathcal{L}_{patchKL}$ of Algorithm 1 and RENOISE additionally uses an autocorrelation loss with $\lambda_{pair} = 10$ (Garibi et al., 2024). Both also employ the reversible inversion trajectory estimation while DDIM does not. The Figures show that both MELODYFLOW and RENOISE outperform DDIM inversion by a large margin on the three evaluated axes. an optimum can be achieved around $\lambda_{KL} = 0.2$ for velocity prediction and around 0.1 for noise prediction. Overall the quality is better (lower FAD$_{edit}$ in the Figure 2a) when directly regularizing the velocity prediction. In both cases we observe a higher CLAP$_{edit}$ in the Figure 2b when the original text description $c_{orig}$ conditions the inversion process, confirming better text-adherence. This happens at the expense of a higher FAD$_{edit}$ compared with unconditional inversion.

#### 4.3.2 TARGET INVERSION FLOW STEP

In the Figures 3a, 3b and 3c we report music editing objective metrics as a function of $T_{edit}$, comparing DDIM inversion with MELODYFLOW. The consistency with the source sample is higher (lower LPAPS) with our method than DDIM inversion. The S-shaped FAD curves of the Figure 3a indicate an inversion optimum around $T_{edit} = 0.06$, correlating with the peak in CLAP$_{edit}$ score.

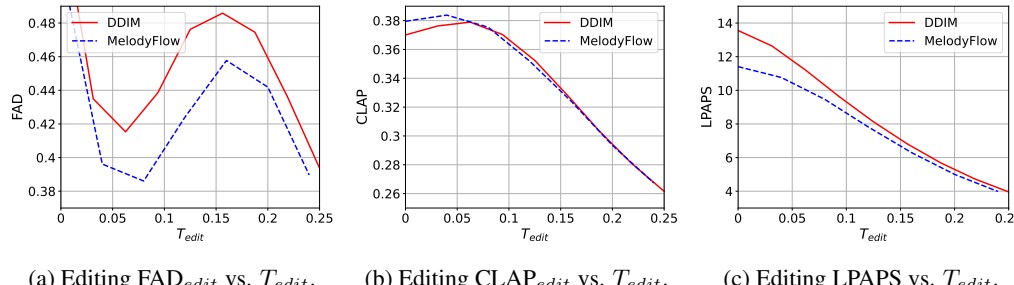

(a) Editing $\text{FAD}_{edit}$ vs. $T_{edit}$.    (b) Editing $\text{CLAP}_{edit}$ vs. $T_{edit}$.    (c) Editing LPAPS vs. $T_{edit}$.

Figure 3: Music editing quality as a function of the target inversion step $T_{edit}$. We report $\text{FAD}_{edit}$ (Figure 3a), $\text{CLAP}_{edit}$ (Figure 3b) and LPAPS (Figure 3c) objective metrics.

Table 3: Codec bottleneck and framerate ablation for 32 kHz mono audio. Both compression and generative model performances are reported on the IN-DOMAIN test set.

| REGULARIZER | FRAME RATE (HZ) | $\text{STFT}_{loss}$ ↓ | SI-SDR↑ | $\text{FAD}_{vgg}$ ↓ |
|---|---|---|---|---|
| $\varnothing$ | 50 | 0.35 | 18.5 | 0.68 |
| RVQ | 50 | 0.55 | 4.4 | 0.55 |
| KL | 50 | 0.34 | 18.1 | 0.48 |
|  | 20 | 0.44 | 12.9 | 0.47 |
|  | 5 | 0.53 | 3.5 | 0.67 |

## 4.4 CODEC BOTTLENECK REGULARIZER

All other things being equal, we ablate on the bottleneck regularizer for a fixed frame rate of 50 Hz by comparing RVQ- (using 4 codebooks of size 2048 each), KL-regularizer (Evans et al., 2024a) and no regularizer at all in the Table 3. Our results indicate optimal reconstruction performance with no regularizer, closely followed by KL. RVQ stands much further away, likely due to the high level of compression enforced by the discretization (despite the significant dictionary size of $2048^4 = 1.7 \times 10^{13}$). The same ranking applies for SI-SDR (Le Roux et al., 2019). Regarding text-to-music generation performance, the KL-regularizer outperforms the other options. Overall this shows the KL-regularizer offers the best trade off between reconstruction and generation performance.

Ablating on the codec frame rate with the KL regularizer shows that 5 Hz achieves comparable performance with the 50 Hz codecs trained with RVQ or no regularizer, a $10\times$ improvement factor. We chose to work with the 20 Hz KL-regularized codec for the 32 kHz mono MELODYFLOW-small, as it offers a good trade off between quality and speed. Accounting for the additional information to compress when scaling to 48 kHz stereo, we chose a frame rate of 25 Hz for MELODYFLOW-medium.

## 4.5 FM DESIGN

### 4.5.1 MODEL TRAINING

We compare our FM model design with the baseline implementation of Le et al. (2024), both being trained on the same music latents. The most notable changes are the removal of the infilling objective during training, the change in flow step sampling and the introduction of mini-batch coupling. Table 4 presents the impact of those choices on the last FM model validation $\text{MSE}_{loss}$ of the EMA checkpoint, and on the in-domain test FAD (in the text-to-music generation setting). No loss value is reported for the baseline as the infilling objective facilitates the task (hence values are not fairly comparable), and for validation we sample flow steps uniformly regardless of the training sampling scheme. Such infilling objective in Le et al. (2024)'s FM model was designed to handle variable length sequences that are inherent to the speech domain. In our experiments it showed to be detrimental for the model performance, and we know diffusion models can support infilling/outfilling without additional tweaks (Liu et al., 2023a). With all methods combined the in-domain FAD is reduced to 0.39 from 0.53 and consistent with the observed loss decrease, which validates our design.

Table 4: FM model design ablation. FAD (resp. MSE) is reported on the IN-DOMAIN test (resp. validation) set. Baseline is adapted from (Le et al., 2024) but retrained on our music latents.

| ABLATION | HEADS | LAYERS | INFILL | SAMPLING | OT-FM | $MSE_{loss}\downarrow$ | $FAD_{vgg}\downarrow$ |
|---|---|---|---|---|---|---|---|
| baseline | 16 | 24 | ✓ | uniform | ✗ | - | .53 |
| − infilling | 16 | 24 | ✗ | uniform | ✗ | .8596 | .50 |
| + sampling | 16 | 24 | ✗ | logit-normal | ✗ | .8484 | .44 |
| + batch coupling | 16 | 24 | ✗ | logit-normal | ✓ | .8322 | .42 |
| + wider model | 18 | 18 | ✗ | logit-normal | ✓ | .8310 | .39 |

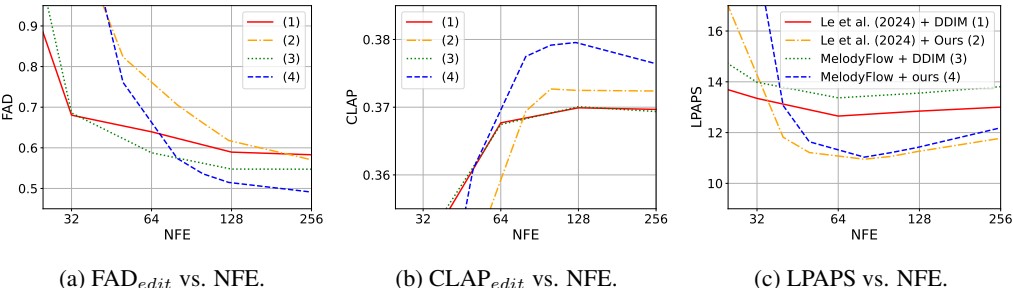

(a) $FAD_{edit}$ vs. NFE.  (b) $CLAP_{edit}$ vs. NFE.  (c) LPAPS vs. NFE.

Figure 4: Efficiency-quality trade offs of MELODYFLOW in the text-guided music editing setting, measured using objective metrics. Objective metrics ($FAD_{edit}$ in the Figure 4a, $CLAP_{edit}$ in the Figure 4b and LPAPS in the Figure 4c) indicate a sweet spot around 128 NFE.

### 4.5.2 MUSIC EDITING QUALITY/EFFICIENCY

We compare DDIM with MELODYFLOW's inversion using a target inversion step of $T_{edit} = 0$. $FAD_{edit}$ (Figure 4a), $CLAP_{edit}$ (Figure 4b) and LPAPS (Figure 4c) are plotted as a function of the total NFE count (inversion + regeneration included). Quality-wise, the combination of our FM and inversion designs outperform the baseline. Regardless of the FM design choice, DDIM inversion requires as few as 32 NFEs to achieve an acceptable FAD. Our inversion only outperforms after 125 NFEs. On the text-adherence axis, the FM model design alone does not translate in better performance when combined with DDIM inversion. Swapping DDIM with our method shows a different trend, highlighting the benefit of combining FM and inversion methods. Analyzing the consistency with the original sample, again we observe that the regularized inversion plays a more important role than the FM model design: the baseline FM model actually outperforms ours when used in conjunction with DDIM inversion. Overall our method consistently outperform the baseline for 125 NFEs.

## 5 RELATED WORK

### 5.1 AUDIO REPRESENTATION

Recent advancements in neural codecs have seen the application of VQ-VAE on raw waveforms, incorporating a RVQ bottleneck as demonstrated in Zeghidour et al. (2021); Défossez et al. (2022), later refined as per Kumar et al. (2024). Evans et al. (2024a) proposed a modification to this approach by replacing the RVQ with a VAE bottleneck to enhance the modeling of continuous representations. In addition, several recent audio generative models have adopted Mel-Spectrogram latent representations, coupled with a vocoder for reconstruction, as shown in the works of (Ghosal et al., 2023; Liu et al., 2023b; Le et al., 2024).

### 5.2 TEXT-TO-MUSIC GENERATION

Models that operate on discrete representation are presented in the works of (Agostinelli et al., 2023; Copet et al., 2024; Ziv et al., 2023). Agostinelli et al. (2023) proposed a representation of music using multiple streams of tokens, which are modeled by a cascade of transformer decoders conditioned on a joint textual-music representation (Huang et al., 2022b). Copet et al. (2024) introduced a single-stage

language model that operates on streams of discrete audio representations, supporting both 32 kHz mono and stereo. Ziv et al. (2023) replaced the language model with a masked generative single-stage non-autoregressive transformer. Schneider et al. (2023); Huang et al. (2023); Liu et al. (2023b) use diffusion models. Schneider et al. (2023) utilized diffusion for both the generation model and the audio representation auto-encoder. Liu et al. (2023b) trained a foundational audio generation model that supports music with latent diffusion, conditioned on autoregressively generated AudioMAE features (Huang et al., 2022a). Evans et al. (2024a;b) proposed an efficient long-form stereo audio generation model based on the latent diffusion of VAE latent representations. This model introduced timing embeddings conditioning to better control the content and length of the generated music.

## 5.3 MUSIC EDITING

Lin et al. (2024) proposed a parameter-efficient fine-tuning method for autoregressive language models to support music inpainting tasks. Garcia et al. (2023) developed a masked acoustic modeling approach for music inpainting, outpainting, continuation and vamping. Wu et al. (2023) fine-tuned a diffusion-based music generation model with melody, dynamics and rhythm conditioning. Novack et al. (2024) is a fine-tuning free framework for controlling pre-trained text-to-music diffusion models at inference-time via initial noise latent optimization. Zhang et al. (2024) investigated zero-shot text-guided music editing with conditional latent space and cross attention maps manipulation. Manor & Michaeli (2024) employs DDPM inversion (Huberman-Spiegelglas et al., 2023) for zero-shot unsupervised and text-guided audio editing.

## 6 DISCUSSION

**Limitations**   The proposed model specifically focuses on text-guided audio editing with the quality/efficiency trade off in mind, hence we do to not aim nor claim to outperform previous state of the art text-to-music generation models. Under our current setup text-guided music editing prompts are not instructions. They describe what the edited sample should sound like given an original music sample and description, but the model is not designed to understand direct editing instructions like *replace instrument A by instrument B*. While we observed that MELODYFLOW performs convincing editing tasks for several axes (genre or instrument swap, tempo modification, key transposition, inpainting/outpainting), more research work is required to accurately evaluate each of those axes. Music editing human listening tests are conducted for a fixed $T_{edit}$, but eventually it should depend on the sound designer's preference on the creativity-consistency axis. Finally the reported objective metrics are mostly used as a proxy for subjective evaluations but they have their limitations. As an example we observe that optimizing FAD for MusicCaps is usually achieved by overfitting on our training dataset, which negatively correlates with perceived quality. Overall subjective evaluations remain the best source of truth until a model that mimics human ratings is developed.

**Conclusion**   In this work we presented MELODYFLOW, the first non-autoregressive model tailored for zero-shot test-time text-guided editing of high-fidelity stereo music. In the text-to-music setting the model offers competitive performance thanks to a low frame rate VAE codec and FM model featuring logit-normal flow step sampling, optimal-transport minibatch coupling and L-shaped attention mask. Combined with our proposed regularized latent inversion method, MELODYFLOW outperforms previous zero-shot test-time methods by a large margin. The model achieves remarkable efficiency that is key for the sound design creative process and supports variable duration samples. Our extensive evaluation, that includes objective metrics and human studies, highlights MELODYFLOW promise for efficient music editing with remarkable consistency, text-adherence and minimal quality degradation compared with the original, while remaining competitive on the task of text-to-music generation. For future work we intend to explore how to accurately evaluate specific editing axes and how such a model could help design metrics that better correlate with human preference.

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

# A   APPENDIX

## A.1   EXPERIMENTAL SETUP

### A.1.1   AUDIO LATENT REPRESENTATION

Our compression model implementation is that of Copet et al. (2024)[3] enhanced by band-wise discriminators and snake activations from Kumar et al. (2024), perceptual weighting (Wright & Välimäki, 2019), VAE bottleneck and multi resolution STFT reconstruction loss from Evans et al. (2024a). We train a mono 32 kHz codec at 20 Hz frame rate and another one supporting stereo 48 kHz audio at 25 Hz. The bottleneck dimension is of 128. Both are trained on one-second random audio crops for 200K steps, with a constant learning rate of 0.0003, AdamW optimizer and loss balancer of (Défossez et al., 2022). Stereo codecs are trained with sum and difference loss (Steinmetz et al., 2020). The bottleneck layer statistics are tracked during training (dimension-wise) for normalization prior to FM model training.

### A.1.2   FLOW MATCHING MODEL

MELODYFLOW's DiT follows Esser et al. (2024) configurations where each head dimension is of 64 and the model has the same number of heads and layers (either 18 or 24). Model implementation is that of `audiocraft`[4] but adapted for FM following Vyas et al. (2023): U-shaped skip connections are added along with linear projections applied after concatenation with each transformer block output Bao et al. (2023). The model is conditioned via cross attention on a T5 representation (Raffel et al., 2020) computed from the text description of the audio, using 20% dropout rate during training in anticipation for the classifier free guidance applied at inference. Cross attention masking is used to properly adapt to the text conditioning sequence length of each sample within a batch and we use zero attention for the model to handle unconditional generation transparently. No prepossessing is applied on the text data and we only rely on the descriptions (additional annotations tags such as musical key, tempo, type of instruments, etc. are discarded, although they also sometimes appear in the text description). The flow timestep is injected following Hatamizadeh et al. (2023). Minibatch coupling is computed with `torch-linear-assignement`[5]. MELODYFLOW-small (resp. MELODYFLOW-medium) is trained on latent representation sequences of 32 kHz mono (resp. 48 kHz stereo) segments of 10 (resp. 30) seconds, encoded at 20 Hz frame rate (resp. 25 Hz). MELODYFLOW-small (resp. MELODYFLOW-medium) is trained for 240k (resp. 120k) steps with AdamW optimizer ($\beta_1 = 0.9$, $\beta_2 = 0.95$, weight decay of 0.1 and gradient clipping at 0.2), a batch size of 576 and a cosine learning rate schedule with 4000 warmup steps. Additionally, we update an exponential moving average of the model weights ever 10 steps with a decay of 0.99. Each model is trained on 8 H100 96GB GPUs with `bfloat16` mixed precision and FSDP (Zhao et al., 2023). MELODYFLOW-small requires 3 days and MELODYFLOW-medium 6 days of training.

### A.1.3   LLM-ASSISTED EDITING PROMPT GENERATION

For editing prompts design we prompted the LLama-3 large language model Dubey et al. (2024) to modify the original descriptions by targeting genre swapping. Edited descriptions were then manually verified to ensure their plausibility and coherence. As an example, given the original description *This is a lush indie-folk song featuring soaring harmony interplay and haunting reverb-y harmonica*, the resulting editing prompt is *This is a lush Indian classical-inspired song featuring soaring harmony interplay and haunting reverb-y bansuri flute*.

### A.1.4   SUBJECTIVE EVALUATION FORM

A screenshot of the music subjective evaluation form is shown in the Figure 5.

---

[3] https://github.com/facebookresearch/audiocraft/blob/main/audiocraft/models/encodec.py

[4] https://github.com/facebookresearch/audiocraft/blob/main/audiocraft/modules/transformer.py

[5] https://github.com/ivan-chai/torch-linear-assignment

Figure 5: Music editing subjective evaluation form. Given the original song A, raters are asked to evaluate three different edits of A, on the three following axes: quality, text adherence, consistency.

## A.2 ADDITIONAL EXPERIMENTS

### A.2.1 STEREO CODEC

The Table 5 reports the impact of scaling from mono to stereo with the same MELODYFLOW-medium model size (1B parameters) trained on 30 second segments. Two codecs are trained on 48 kHz audio, using the same 25 Hz latent frame rate: one mono and one stereo. The in-domain FAD is reported for 10s and 30s generated segments. Moving from mono to stereo marginally affects the generative model performance.

### A.2.2 LENGTH GENERALIZATION

One drawback of training the FM model of a fixed segment duration is that the inference can only be run for the same duration, otherwise the quality will degrade (this can be seen in the $FAD_{10s}$ column of the Table 5, when comparing the first two rows). This can be handled by using padded segments and specific conditioning (Evans et al., 2024a), but does not save any resource when targeting shorter segments. Another solution is to train on variable length segments but then the model does not generalize well for full length segments, and will better learn for the uttermost left positions of the sequence that appear more often. We propose to simulate training on variable length segments, while keeping the model learning for the full length scenario. This is done by applying a L-shaped attention mask during model. For each sequence of length L, we randomly select a segment boundary within the range $[0, L]$. Positions before the boundary can only attend to themselves in the DiT's self-attention, while positions after it attend to the entire sequence.

Comparing the first two rows of the Table 5 indicate that the L-shaped mask helps supporting versatile duration with no penalty on full-length segments, unlocking faster inference for segments shorter than 30 seconds. This method does not generalize to segments longer than 30 seconds, which should be specifically handled with a sliding window/outpainting approach.

Table 5: Ablation on L-shaped attention mask and stereo for MELODYFLOW-large. Each variant is trained on 30s audio segments encoded with a 25 Hz frame rate codec trained on 48 kHz audio.

| CHANNELS | STFT$_{loss}$ ↓ | SI-SDR↑ | L-MASK | FAD$_{10s}$ ↓ | FAD$_{30s}$ ↓ |
|---|---|---|---|---|---|
| 2 | 0.40 | 12.48 | ✓ | 0.59 | 0.65 |
| | | | ✗ | 1.48 | 0.65 |
| 1 | 0.39 | 13.34 | ✓ | 0.49 | 0.59 |

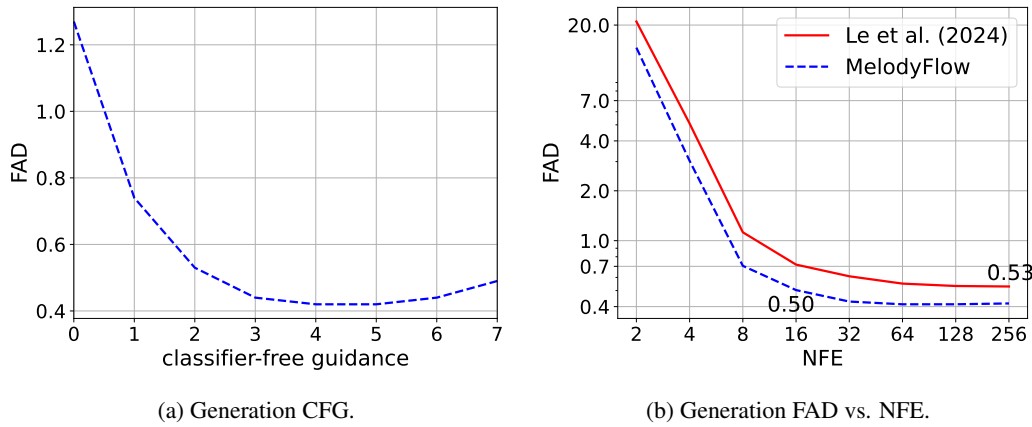

(a) Generation CFG.          (b) Generation FAD vs. NFE.

Figure 6: Text-to-music generation quality (FAD) as a function of classifier-free guidance factor (Figure 6a) and inference steps (Figure 6b). The baseline of the Figure 6b is the FM model architecture of (Le et al., 2024) but retrained on our music latents. The combination of our flow matching design choices enable faster generation for a given efficiency budget or better overall quality.

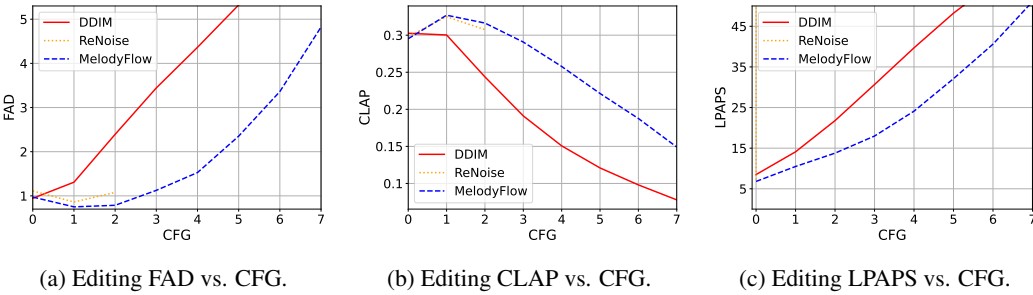

(a) Editing FAD vs. CFG.   (b) Editing CLAP vs. CFG.   (c) Editing LPAPS vs. CFG.

Figure 7: Music editing objective metrics as a function of the classifier free guidance, using the same CFG for both inversion and regeneration.

### A.2.3 TEXT-TO-MUSIC GENERATION EFFICIENCY

In the Figure 6b we report the text-to-music generation test FAD as a function of the number of DiT forward passes (NFEs) for both the baseline FM architecture (Le et al. (2024)) and final version of MELODYFLOW. Not only does MELODYFLOW achieve better performance, but with 16 times fewer NFEs (e.g. where the baseline required 256 NFEs to reach 0.53 FAD, MELODYFLOW only requires 16 NFEs to score 0.50).

### A.2.4 CLASSIFIER-FREE GUIDANCE

In the Figure 6a we report the in-domain test FAD as a function of the classifier-free guidance factor in the text-to-music generation setting. We use a classifier-free guidance factor of $4.0$ for the text-to-music generation inference.

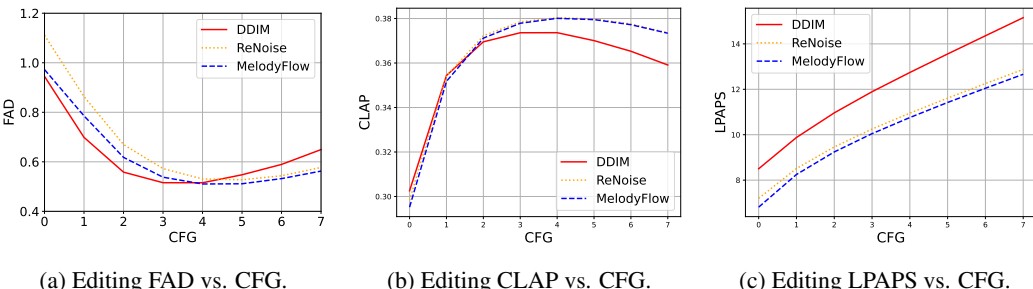

(a) Editing FAD vs. CFG.    (b) Editing CLAP vs. CFG.    (c) Editing LPAPS vs. CFG.

Figure 8: Music editing objective metrics as a function of the classifier free guidance, when using a CFG of 0 during latent inversion.

In the Figures 7a, 7b, 7c we plot our objective metrics for text-guided music editing, as a function of the CFG. The performance is bad whatever the considered inversion method, showing that using the CFG during inversion is detrimental. Above a CFG of 0 RENOISE completely diverges (the LPAPS skyrockets), while MELODYFLOW achieves the best robustness. This explains why we consider that RENOISE is not directly compatible with the FM formulation, even after converting to $\epsilon$-prediction, and that FM requires an adapted regularized latent inversion method. After keeping the CFG to zero during latent inversion to stabilize the process, the results are presented in the Figures 8a, 8b, 8c. We end up using the same classifier-free guidance factor of $4.0$ for our editing experiments.

