# OpenReview forum: "High Fidelity Text-Guided Music Editing via Single-Stage Flow Matching"
_ICLR.cc/2025/Conference — ICLR 2025 Conference Withdrawn Submission_

### Official Review · Reviewer_tg1s · 2024-10-30

**Soundness:** 2
**Presentation:** 3
**Contribution:** 1
**Rating:** 3
**Confidence:** 4

**Summary:**

This paper leverages the diffusion transformer architecture with flow matching objective for text-to-music generation and editing at 48kHz stereo quality. The model adopts a variational autoencoder to compress the raw stereo waveforms to latent representations. Subsequently, a flow matching model is trained for text-based conditional generation. For the editing component, the authors adapt an optimization-based inversion method from ReNoise, integrating it into the flow matching framework. The experimental results demonstrate that the proposed text-based editing approach surpasses both DDIM Inversion and ReNoise on objective metrics.

**Strengths:**

1. This paper introduces a novel single-stage flow matching model for text-to-music generation, capable of generating and editing audio samples at 48 kHz stereo quality.
2. The authors exploit a regularized flow matching inversion method to facilitate text-based music editing and conduct ablation studies to validate its effectiveness.
3. The experimental results demonstrate that the proposed approach outperforms all baseline methods across all objective metrics.

**Weaknesses:**

1. Although the paper claims to introduce the flow matching model for text-to-music generation, it is apparent that its performance on subjective metrics does not match that of Stable Audio. Furthermore, the authors have not provided an evaluation of Stable Audio's objective metrics.

2. Need a more comprehensive literature review. Important editing methods like DITTO or MEDIC are not in the literature review. Also need to compare with MusicMagus in zero-shot editing.

3. This work proposes a new regularized inversion method; however, the primary distinction from ReNoise lies in the alteration of noise prediction to velocity prediction. While the results achieved are commendable, the contribution lacks sufficient novelty.

4. The authors assert that their method retains similarity to the source audio; however, it obtains a lower score on the CON metric in subjective evaluations compared to AudioLDM2-Music. Moreover, the demonstrated demo reveals inadequate preservation of essential content, raising concerns about the validity of their claim regarding the maintenance of the original audio's critical elements.

5. The paper claims capability of handling 30 seconds of music, yet it predominantly conducts comparative experiments using MusicCaps, which features a duration of only 10 seconds. It is advisable for the authors to include comparisons with longer music datasets, such as the Song Describer Dataset.

6. The authors have not yet made their code and model publicly available, which should not be presented as a contribution point in the paper.

**Questions:**

See questions in weakness.

---

### Official Review · Reviewer_aHmV · 2024-11-01

**Soundness:** 3
**Presentation:** 3
**Contribution:** 2
**Rating:** 5
**Confidence:** 3

**Summary:**

This paper presents MelodyFlow, a novel Flow-Matching based TTM model capable of generating 30 seconds of 48 kHz audio, and in parallel introduce a regularized inversion method to allow for inference-time editing.

**Strengths:**

- Overall writing style is clear
- Introduction of FM objective is streamlined and easy to follow.
- The breadth of ablations are much appreciated, as the authors go to reasonable lengths to understand the design space and limitations of their method.

**Weaknesses:**

- Line (046-047) “editing methods from the computer vision domain, which are exclusive to diffusion models (Novack et al., 2024; Zhang et al., 2024; Manor & Michaeli, 2024)” is not wholly true. Optimization methods like Novack et al., 2024 are agnostic to the sampling process (and in fact, the flow-matching equivalent has already been explored [1]), and guidance methods like Zhang et al., 2024 are also agnostic to sampler (as nothing specific about their method *requires* DDIM inversion as the inversion method).
- In general, Section 2.4 is not clear enough. It is hard to tell which parts of the the algorithm from ReNoise are being kept and which are novel, and in general as there is very little textual explanation of the method (and only consistent referrals back to the ReNoise paper), it is hard to tell where the novelty lies. As this is a claimed **core contribution** of the paper, more time should be spent in section 2.4 to make the differences between the baseline and their method explicitly clear, and to explain the algorithm in depth textually. Along with this, Figure 1 is very hard to parse, and a more semantic diagram (rather than ground truth control flow) may be more helpful.
- 2.5.2 is also very much understated, and is hard to parse without an intimate knowledge of the references. As the paper uses this method to show its increased performance, it should be made explicit in the paper how this works and what the goal of the method is (e.g. is this a training or inference time change? Why do we want to run the Hungarian algorithm?)
- I wonder at a high level that the overall framing (and title) of the paper is somewhat misplaced. On line 047, it is stated that “Despite recent efforts, no approach has yet shown the ability to perform high-fidelity generic style transfer across various music editing tasks,” which I think is the right framing for the inversion task used here, as the example samples show much more strengths in “stylistic” changes rather than grounded content changes (i.e. they very much sound “inspired” by the original samples rather than editing them). This is a fine approach for the paper’s framing, but this is the *only* mention of style in the entire paper, and thus much of the paper sounds like its talking about some other, more grounded form of editing.
- While previous works have used it, it is publicly documented that the VGGish backbone for FAD is not correlated with human perception [2], and thus some other backbone (like CLAP) should be used. This shouldn’t be too much of an issue, as all the models tested here are open-source (with the exception of Stable Audio, but Stable Audio Open exists) to run directly.
- Overall, the technical results are somewhat weak. While it is mentioned that in the limitations the goal is music editing and *not* to outperform other TTM models (line 511), this is at odds with their first main stated contribution of making an improved TTM generation model *in general* (line 097). The results against existing baselines show no clear strengths other than moderate latency speed ups.  It also seems like between the baseline ReNoise and the proposed method, the differences are small.

As a whole, the paper seems to be caught somewhere between being a modeling paper (in its proposal of ways to improve training of FM-based TTM models) and a music editing paper (through its FM inversion method), weakening it in both respects. The strongest modeling results are only with respect to an internal non-TTM baseline and do not seem to get comparable results with current SOTA methods, and a full ablation of FM TTM modeling is given to the space dedicated to the editing results. And conversely, without the modeling contributions, it seems like the core contribution then rests solely on the FM inversion technique, in which its novelty is hard to assess. I do not think the present version of the paper is ready for acceptance, though I think changes to spending more time on the editing results (as those seem comparatively stronger than the modeling changes) as the main contribution and overall framing would improve the paper substantially.

[1] Ben-Hamu, Heli et al. “D-Flow: Differentiating through Flows for Controlled Generation.” ArXiv abs/2402.14017 (2024): n. pag.

[2] Gui, Azalea et al. “Adapting Frechet Audio Distance for Generative Music Evaluation.” ICASSP 2024 - 2024 IEEE International Conference on Acoustics, Speech and Signal Processing (ICASSP) (2023): 1331-1335.

**Questions:**

- “MELODYFLOW uses a DiT of sizes 400M (small) and 1B (medium) parameters with U-shaped skip connections Bao et al. (2023).” Is this not just a U-Net then? As many TTA/TTM UNets utilize internal attention layers, it’s unclear whether this is an actual DiT or some DiT-UNet hybrid (though I may be misunderstanding something here).
- For figure 3, what happens after $T_{edit} > .25$? Especially in the case of FAD, it seems like another optimum may arise if $T_{edit}$ is increased further.

---

### Official Review · Reviewer_9va9 · 2024-11-02

**Soundness:** 3
**Presentation:** 3
**Contribution:** 2
**Rating:** 6
**Confidence:** 4

**Summary:**

This work proposes a text-to-music model based on a DiT(diffusion transformer) established on the continuous latent of VAE, in the setup of flow matching. This model not only supports usual generation conditioned by text, but also music editing by depicting music editing as the combination of latent inversion plus re-generation with a new prompt. During the latent inversion, a KL regularization method derived from ReNoise is introduced to enhance the quality and coherence to the original sample.

This work incorporated several techniques for quality improvements and shown their usefulness in ablation studies (Table 3, 4). Other ablations shown the trend of changing NFEs, lambda_KL and T_edit (Fig. 2,3,4). Furthermore, the ablations and experiments show that, when proper hyperparameters are chosen, the regularized latent inversion can outperform DDIM and ReNoise. The quality on generation and text-guided editing of proposed work is evaluated by both subjective tests and objective metrics.

**Strengths:**

- This work is adapts ReNoise with flow matching on music modality plus its own modification, which demonstrates the feasibility of such methodology on music editing. The objective and subjective results also shown that the proposed model have better performance on editing tasks compared to other baselines.

- In terms of generation, proposed model is capable of generating samples which have quality on par with baseline models in a much shorter inference latency.

- The proposed model and method are well presented. Design choices in autoencoder and training techniques are well-described. Moreover, their usefulness are justified with ablation studies.

- Authors plan to release model weights and code, which will benefit the community for sure.

- The limitation part clearly mentioned several issues and limitations of the proposed work, which is also insightful for readers.

**Weaknesses:**

- Components of this work is a combination from existing works. Its novelty lies within the modification to the regularization method of ReNoise, but it lacks theoretical support/analysis on the effectiveness of such modification. It should be worth to have more discussion and analysis on, for example, why removing L_pair improves the result, whether there's a sweetspot for lambda_pair, and what's the side effect of removing L_pair. Empirically, it might be helpful to have a grid search based on both lambda_pair and lambda_KL, and measure the KL-divergence between the latents from different approaches. Visualizing the distribution of the latents with low dimensional projection such as t-SNE or UMAP could also be interesting.

- The quality evaluation could not show whether the proposed model is qualified to be called as "high-fidelity". Although the the FAD score is the lowest for the proposed model, but FAD_vgg cannot evaluate the content beyond 16k due to the limitation of the VGG model. The experiment included MusicCaps as the reference set in the subjective evaluation, but the audio of MusicCaps is a subset of AudioSet, which is obtained from Youtube video, thus unlikely to be in high-fidelity format(uncompressed 44.1 or 48 khz). Moreover, the subjective test result on overall quality does not favor proposed work (around the same as MusicGen-medium, but not at the same level of StableAudio). One way to justify this is to compare the generated samples of the proposed model with another reference set which is natively 48khz. A held out set of the internal 48khz dataset could serve this purpose.

**Questions:**

- It would be great to know how what's the composition of the in-domain evaluation set. For example. does it curated to have balanced genres like the set used in subjective tests? An example of such breakdown can be found in the Appendix A of [1].
[1] MusicLM: Generating Music From Text, Andrea Agostinelli, Timo I. Denk, Zalán Borsos, Jesse Engel, Mauro Verzetti, Antoine Caillon, Qingqing Huang, Aren Jansen, Adam Roberts, Marco Tagliasacchi, Matt Sharifi, Neil Zeghidour, Christian Frank,
URL: https://arxiv.org/pdf/2301.11325

- Will the best value for lambda_KL dependent on specific network architecture or even dependent with network size?

- The problem setup of text-guided editing in this work has an implicit assumption: The coherence to x_src can be achieved with ODESolve(0->t_edit), and the adherence of the editing can be achieved with ODESolve(t_edit->1). I understand this is an assumption existed from previous works, but it lacks evidence that it can always hold for most editing scenarios. Could it be possible to categorize the text prompts into those categories mentioned in L042-043 and L514-515 and show the categorical-wise editing performance?

Heuristically, during the forward diffusion, timbre information is more likely to be erased before the note information. Therefore, an interesting case to test about this assumption would be find a way to promp the model to change the melody(e.g. raise the melody with 3 semitones) but maintaining the timbre on an audio that has predominant melody trajectory. Conventional MIR algorithms/metrics can be used to evaluate the result.

- I am not sure if I am over cautious, but since the model checkpoint will be released, it would be nice to ensure if all the licenses of training sets are compatible with this.

---

### Official Review · Reviewer_Rept · 2024-11-02

**Soundness:** 2
**Presentation:** 3
**Contribution:** 2
**Rating:** 3
**Confidence:** 4

**Summary:**

The authors propose MelodyFlow, a flow-matching-based (FM-based) text-to-music generative model capable of producing 48kHz, stereo-format music. Additionally, they introduce a music editing method built on the pretrained MelodyFlow. Specifically, they adapt diffusion model-based inversion methods, such as DDIM inversion and ReNoise, to work with FM-based models, enabling editing through FM-based inversion. They experimentally demonstrate MelodyFlow's capabilities in both text-to-music generation and music editing.

**Strengths:**

- Music generation: The authors experimentally demonstrate that their FM-based text-to-music generative model can produce 48kHz, stereo-format waveforms. To achieve this, as discussed in Section 2.5, Appendix A.2, and Table 4, they conduct a detailed exploration of architectural improvements, including audio compression parts and enhanced training techniques such as Minibatch Coupling.
- Music editing: They reformulate techniques of improved DDIM inversion, as proposed in Pix2pix-zero [1] and ReNoise, to make them applicable to FM-based music generative models. A key point in the originality of this work lies in not merely applying DDIM inversion or ReNoise directly but specifically refining the regularization technique in ReNoise.

[1] Parmar, G., Kumar Singh, K., Zhang, R., Li, Y., Lu, J. and Zhu, J.Y., 2023, July. Zero-shot image-to-image translation. In ACM SIGGRAPH 2023 Conference Proceedings (pp. 1-11).

**Weaknesses:**

**Overall**:
1. Music generation part:
- The applicability of FM-based generative models for audio data is already explored in some prior work (for example, Text-to-Music Generation [1] and Text-to-Audio [2]). Therefore, simply applying FM-based model to text-to-music generation task (even 'single-stage') does not contribute significantly to new insights for the audio/music community.
    - Therefore, additional novel technical contributions, such as sophisticated ideas to surpass the sample quality of existing text-to-music models (regardless of whether the FM-based or diffusion-based models) or techniques that enable more efficient generation while maintaining sample quality, along with sufficient experimental evaluation to demonstrate their effectiveness, should be explored.

- Considering of above, the authors' efforts to improve generation performance through architectural and training enhancements (as discussed in Section 2.5, Appendix A.2, and Table 4) could still provide some valuable insights. However, even from this perspective, both objective and subjective metrics in Table 2 suggest that, despite incorporating various techniques, the main advantage of the FM-based models lies in halving the number of sampling steps compared to diffusion-based models like Stable Audio. Thus, the contributions offered from this angle remain limited.

2. Music-editing part:
- From the perspective of ODE-based generative models, score-based and FM-based generative models share a broadly similar framework. Thus, the fundamental contribution of reformulating DDIM inversion or its extensions, such as Pix2pix-zero/ReNoise, to apply to FM-based models is not particularly significant. (Of course, I acknowledge the practical differences, so please note that I am not evaluating the proposed editing method solely from this perspective.)
- Regarding the proposed inversion technique, the experimental results do not seem to fully support its effectiveness. For example, in Table 1, considering the inherent challenges of subjective evaluation in music editing tasks and the CI95, the advantage over DDPM inversion in terms of 'AVG.' does not appear sufficiently clear. (To clarify, I am indicating that the results are not clear enough; I am not suggesting that the proposed method is ineffective. For more details on objective metrics, please refer to Details below and Questions 1~3.)

**Details**:

The objective metrics used for music editing and generation leave some uncertainty regarding their adequacy in validating the effectiveness of the proposed methods.

- Recent studies, such as [3], suggest that using FAD with VGG-ish to evaluate music samples is inappropriate. To better support the authors' claims in this work, I recommend using alternative pretrained models as introduced in [3] (for both editing and generation tasks).
    - For example, based on the correlation between MOS and FAD discussed in [3], it would be more appropriate to use FAD with LAION-CLAP to evaluate musical quality, and to use FAD with DAC/EnCodec embeddings to assess acoustic quality (please see more detail in [3]).
- For LPAPS, if the authors used the pretrained network from this repository [4], it was trained on the VGGSound dataset [5] (also, please review its data filtering process). This raises concerns about its validity for numerical evaluation in music editing. Additionally, the checkpoint provided in that repository [4] has other issues. As noted in this issue [6], the authors of this repository acknowledge a problem in the LPAPS training procedure itself and do not recommend using LPAPS for at least training purposes.
    - It would be appropriate to calculate the L2 distance using other audio encoders trained properly. For instance, as in [3], I recommend calculating the L2 distance based on embeddings from audio encoders like LAION-CLAP, DAC, or EnCodec.
- Considering both these points and the subjective evaluation results, the current experimental results do not seem sufficient to demonstrate the effectiveness of the proposed latent KL regularization, which contributes to the originality of this method. I recommend revisiting the evaluation protocol to provide stronger support for contributions of this paper. (I do acknowledge that the authors have demonstrated that FM-based music generative models can achieve music editing through inversion techniques to some extent.)
   -  For instance, conducting additional evaluations using the objective metrics mentioned above, as well as a subjective evaluation for an ablation study on the necessity of latent KL regularization in MelodyFlow’s inversion ($\lambda_{KL}=0$ or not), would support the contribution much more in terms of the proposed editing method if their effectiveness is sufficiently demonstrated.

[1] Tal, O., Ziv, A., Gat, I., Kreuk, F., Adi, Y., 2024 Joint Audio and Symbolic Conditioning for Temporally Controlled Text-to-Music Generation. The 25th International Society for Music Information Retrieval (ISMIR) Conference.

[2] Vyas, A., Shi, B., Le, M., Tjandra, A., Wu, Y.C., Guo, B., Zhang, J., Zhang, X., Adkins, R., Ngan, W. and Wang, J., 2023. Audiobox: Unified audio generation with natural language prompts. arXiv preprint arXiv:2312.15821.

[3] Gui, A., Gamper, H., Braun, S. and Emmanouilidou, D., 2024, April. Adapting frechet audio distance for generative music evaluation. In ICASSP 2024-2024 IEEE International Conference on Acoustics, Speech and Signal Processing (ICASSP) (pp. 1331-1335). IEEE.
Here is toolkit https://github.com/microsoft/fadtk.

[4] https://github.com/v-iashin/SpecVQGAN

[5] Chen, H., Xie, W., Vedaldi, A. and Zisserman, A., 2020, May. Vggsound: A large-scale audio-visual dataset. In ICASSP 2020-2020 IEEE International Conference on Acoustics, Speech and Signal Processing (ICASSP) (pp. 721-725). IEEE.

[6] https://github.com/v-iashin/SpecVQGAN/issues/13

**Questions:**

Questions (see also Weakness):
1. Which checkpoint was used for the CLAP score? There are several checkpoints available at https://github.com/LAION-AI/CLAP.
2. Regarding the subjective evaluation for the music editing task in Table 1, the methods are rated on a 1–5 scale, with none of the methods achieving a rating of 3 or higher. Does this imply that none of the methods achieved a "fair" level of quality according to the standard MOS scale?
3. In Section 4.3.2, L376, the authors claim that LPAPS is being used to measure 'consistency' between source and edited samples. Could you provide at least an intuitive explanation, if not a theoretical one, supporting why LPAPS is suitable for evaluating "consistency"? (Also, could you provide an intuitive definition of "consistency" in this context for better understanding?)

Minor comments:
- Abstract, L16: The term "RENOISE" is not defined.
- Equations 2.2 and 2.3: The equations are missing reference numbers, such as (2.2) and (2.3).

---

### Official Review · Reviewer_r15N · 2024-11-04

**Soundness:** 4
**Presentation:** 3
**Contribution:** 2
**Rating:** 3
**Confidence:** 3

**Summary:**

The authors proposed a method for the music editing task that adapts the ReNoise regularization approach to flow matching. The paper also offer insights on training audio flow matching models, such as Codec bottleneck size and the use of minibatch coupling, which could benefit the community. While the authors provide extensive experimental insights, the overall results and contributions of the paper remain questionable.

**Strengths:**

1- The authors demonstrated good music generation results

2- The authors promised the release of the code and model checkpoints which can be useful for the community

3- The paper is well written.

**Weaknesses:**

- Questionable quantitative results: In Table 1, the proposed method shows marginal improvement in relevance to the editing prompt, but it falls behind in terms of consistency. Additionally, a better overall sound quality does not necessarily indicate improved editing capability. Overall, the metrics presented regarding the editting are not convincing.

- Poor qualitative results: The provided samples in the supp. do not demonstrate fine-grained editing capabilities, as they fail to showcase cases where only one or two words are changed in the prompts. This makes it difficult to assess the quality of the editing. Personally, I found the samples to lack consistency with the source music clips.

- Limited novelty: Although the authors offer valuable insights into training audio FM models, the proposed editing method is primarily an adaptation of the ReNoise technique, which limits the work’s technical novelty.

**Questions:**

- Performance in Table 2: Why does the medium model in Table 2 perform worse than the small model across most metrics?
-  Figure 2: I found Figure 2 is difficult to interpret. What do the four lines correspond to? Additionally, the three subplots report different metrics (FAD, CLAP, and LPAPS), making it impossible to directly compare the three methods under consideration.
- What KL regularization factor was used when training the Codec?

---

### Note · Authors · 2024-11-18

I have read and agree with the venue's withdrawal policy on behalf of myself and my co-authors.